# UniSVQ: 2-bit Unified Scalar-Vector Quantization

**Haoyu Wang** [1]  **Haiyan Zhao**[†][1]  **Xingyu Yu** [2]  **Zhangyang Yao** [3]  **Xu Han**[†][1]  **Zhiyuan Liu** [1]  **Maosong Sun** [1]

## Abstract

Post-training quantization at the 2-bit level enables low-cost deployment and inference acceleration for large language models (LLMs). Scalar quantization (SQ) and vector quantization (VQ) are two primary quantization methods, however, the former suffers from significant performance degradation, and the latter incurs computational and storage overhead. We propose UniSVQ, a unified 2-bit quantization framework that bridges scalar and vector quantization by parameterizing codewords as an affine transform of integer lattices. This structure preserves compatibility with optimized integer kernels while retaining much of VQ's flexibility. We further introduce a data-driven block-wise fine-tuning strategy to directly minimize quantization reconstruction error. Extensive experiments across multiple LLM families and zero-shot benchmarks demonstrate that UniSVQ consistently outperforms state-of-the-art SQ methods and achieves performance comparable to advanced VQ methods, while providing higher inference throughput.

## 1. Introduction

Large language models (LLMs) require substantial computational resources, which creates a barrier to their real-world applications. Various model compression techniques, such as quantization (Frantar et al., 2022; Lin et al., 2024), pruning (Sun et al., 2023; Ma et al., 2023), knowledge distillation (Gu et al., 2023; Wang et al., 2020), and matrix decomposition (Qinsi et al., 2024; Wang et al., 2025), have been adopted to address this challenge and compress large-scale models. Post-training quantization (PTQ) is one of the main quantization approaches. Due to its lower computational cost and minimal performance degradation, PTQ is currently widely used in LLM compression (Hao et al., 2025).

As quantization methods have improved, the performance degradation of PTQ at 4 bits or higher has become relatively minor (Xing et al., 2025; Liu et al., 2025a). Recently, research has begun to focus on extremely low-bit quantization at 2 bits or below (Chee et al., 2023; Tseng et al., 2024a; Baalen et al., 2024; Egiazarian et al., 2024), and these methods can be classified as scalar or vector quantization.

Scalar quantization (SQ) converts each floating-point weight into a finite set of discrete values. The quantization-dequantization processes of SQ are rather simple. Furthermore, well-optimized tensor cores can substantially increase the computational speed of SQ models (Frantar et al., 2025). However, traditional 2-bit SQ methods typically experience severe performance degradation. SQ offers limited flexibility for fine-tuning, and the min-max projection and independent processing of each dimension make SQ susceptible to outliers. Consequently, the performance degradation of state-of-the-art (SOTA) 2-bit scalar-quantized models can exceed 30% on various zero-shot tasks (Liu et al., 2025b).

Vector Quantization (VQ), on the other hand, quantizes groups of contiguous weights together. It maps fixed-length, floating-point weights to one of a finite set of floating-point vectors. This set of vectors is called a codebook, and each vector in the codebook is called a codeword. VQ exhibits superior performance at 2 bits (Egiazarian et al., 2024). However, it requires additional storage for the codebook. If the size of the codebook exceeds the GPU's L1 cache, frequently transferring the codebook between the GPU's memory and the L1 cache can significantly slow down computation (Tseng et al., 2024b). Thus, many VQ studies focus on reducing the codebook size, which usually leads to decreased performance or a more complex decoding process (Egiazarian et al., 2024; Tseng et al., 2024a).

In this work, we show that the unstructured codebook is actually the main cause of the computational and storage overhead of VQ methods. Based on this insight, this paper proposes UniSVQ, a 2-bit quantization method that leverages the advantages of SQ and VQ, aiming to minimize performance degradation while reducing codebook storage overhead and decoding complexity. The key insight is the

---

† Corresponding author [1]Department of Computer Science and Technology, Tsinghua University, Beijing, China [2]School of Software and Microelectronics, Peking University, Beijing, China [3]School of Physics, Beihang University, Beijing, China. Correspondence to: Xu Han <han-xu@tsinghua.edu.cn>, Haiyan Zhao <zhao_haiyan@foxmail.com>.

*Proceedings of the $43^{rd}$ International Conference on Machine Learning*, Seoul, South Korea. PMLR 306, 2026. Copyright 2026 by the author(s).

spatial structure of the quantization grid, i.e., the set of all possible values in the quantized weight matrix. When using a linear-constrained quantization grid, where all discrete values can be obtained via an affine transformation of a group of integer-coordinate vectors, a model structure that lies between scalar and vector quantization can be obtained. During quantization, UniSVQ is equivalent to VQ using a linear-constrained codebook and can achieve similar performance. On the other hand, as shown in Figure 1, UniSVQ replaces the VQ codebook with an affine transformation, reducing the number of additional parameters, and maintains a structure similar to SQ. This enables the reuse of well-optimized SQ matrix multiplication kernels. Our experiments demonstrate that UniSVQ achieves performance superior to SOTA SQ methods and comparable to or better than VQ methods. Additionally, we introduce an adaptive fine-tuning strategy for the affine transformation that directly minimizes the quantization objective function in a data-driven manner. This further improves the quantized model's performance across a wide range of tasks.

In summary, this paper's contributions are as follows:

- We introduce UniSVQ, a 2-bit quantization framework that bridges scalar and vector quantization via an affine lattice parameterization, achieving VQ-like flexibility with only a small number of auxiliary parameters.

- We show that UniSVQ consistently outperforms SoTA SQ baselines and is competitive with strong VQ methods across models and zero-shot benchmarks.

- We demonstrate that UniSVQ improves inference efficiency by reducing codebook-related memory traffic, leading to higher throughput in practice.

## 2. Background and Related Work

This section begins with a review of representative works of vector and scalar PTQ. We then introduce the concepts of UniSVQ and analyze the connections among these methods.

### 2.1. Weight-Only PTQ

PTQ directly converts the weights of a pre-trained model into a low-bit representation. This paper focuses primarily on weight-only PTQ, in which only the weight matrices are quantized. The objective is to minimize the discrepancy of activations after applying the quantization-dequantization function $\Phi()$. For each linear projection, let the input activations be $X \in \mathbb{R}^{N \times n}$, the weight matrix be $W \in \mathbb{R}^{n \times m}$ and the output be $Y = XW^T$. We have

$$\mathcal{L}(\Phi) = \left\| XW^T - X\Phi(W^T) \right\|_2^2. \tag{1}$$

Equation (1) is the same for vector and scalar quantization, and the primary distinction lies in the quantization grids.

### 2.2. Scalar Quantization

SQ methods treat each weight individually, where the quantization result is determined by the weight's value. For the integer quantization considered in this paper, the function $\Phi(\cdot)$ can be defined in a point-wise way:

$$\Phi_{\text{SQ}}(w) = s \cdot (\text{clamp}(\lceil \frac{w}{s} \rfloor + z, q_{\text{min}}, q_{\text{max}}) - z). \tag{2}$$

In Equation (2), $s$ and $z$ denote the scaling factor and the zero point, which are typically shared across the entire weight matrix or several contiguous columns, referred to as a group. $q_{\text{min}}$ and $q_{\text{max}}$ denote the minimum and maximum representable integer values for a given bit width.

SQ is the earliest PTQ method for LLMs. While there is minimal performance degradation above 4 bits (Shao et al., 2024; Ashkboos et al., 2024), typical SQ methods like GPTQ struggle as the bit width decreases (Kumar et al., 2025). In 2-bit quantization, $\Phi_{\text{SQ}}(w)$ within a group can only take 4 distinct quantization values, so selecting these values becomes critical. To address this issue, methods such as OmniQuant (Shao et al., 2024) and SignRound (Cheng et al., 2024) use data-driven approaches to determine suitable values by treating $s$ and $z$ as trainable parameters.

Other works focus on outliers in weight matrices. Previous research indicates that outliers are the primary cause of performance degradation in low-bit quantization (An et al., 2025). The magnitudes of these outliers far exceed those of other values. When quantization grids are set in regions of high probability density, outliers incur significant clipping errors, which greatly reduce model performance. To solve this, SqueezeLLM (Kim et al., 2024) and ICQuant (Li et al., 2025) represent outliers as additional sparse matrices. Other studies, such as PB-LLM (Yuan et al., 2024) and Bi-LLM (Huang et al., 2024), have confirmed that outlier-aware quantization achieves reasonable performance even below 2-bit. These studies show that, when weight distributions are concentrated, 2-bit quantization can retain substantial model capability with only 4 distinct values. Additional evidence for this assertion comes from quantization methods based on orthogonal transformations. For example, QuIP (Chee et al., 2023) adjusts weights to a Gaussian distribution via orthogonal transformations, thereby eliminating the impact of outliers. Follow-up work such as SpinQuant (Liu et al., 2025a), FlatQuant (Sun et al., 2025), and OSTQuant (Xing et al., 2025) further reduces performance degradation by optimizing orthogonal matrices through learnable approaches.

However, despite these advancements, such methods can still result in a performance drop exceeding 50% on challenging tasks, particularly for smaller models. Conversely, vector quantization methods have demonstrated greater potential in 2-bit quantization.

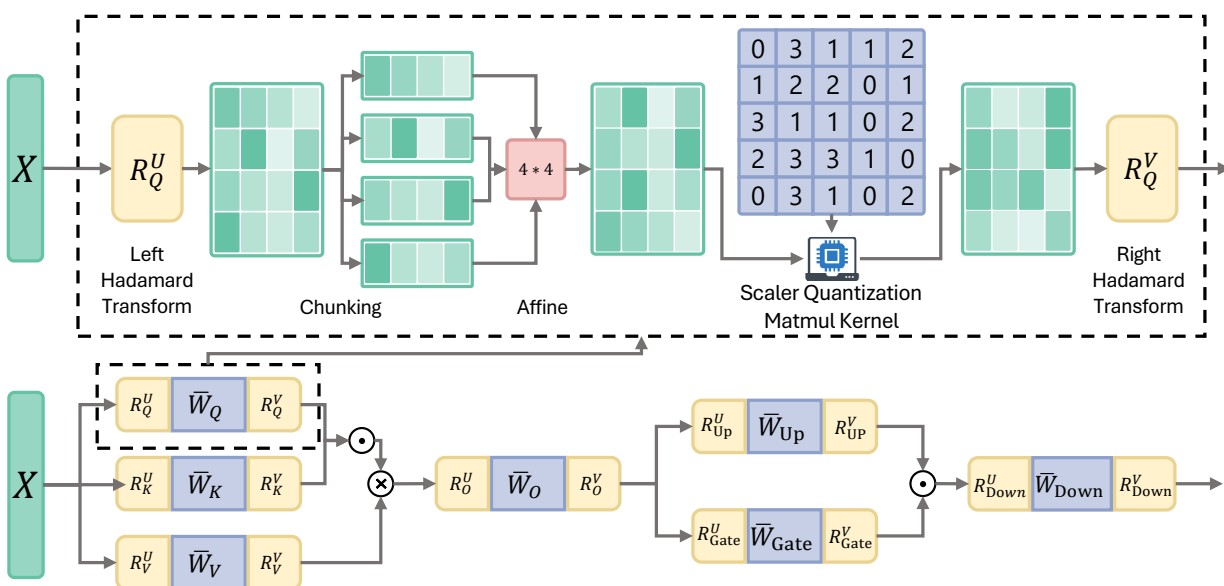

*Figure 1.* Architecture of the UniSVQ method. UniSVQ introduces only 20 additional parameters ($4 \times 4$ for the affine matrix and 4 for the bias vector) per weight matrix, which are significantly fewer than VQ. Besides, the block-wise affine transformation can be pre-applied to the activations, enabling the reuse of well-optimized scalar quantization Matmul kernels. The operation $R_Q^U = U S_U$ denotes the inverse Randomized Hadamard Transform, utilized to suppress outliers and ensure a more uniform weight distribution.

## 2.3. Vector Quantization

VQ quantizes a contiguous group of weights and represents them as a specific codeword from a fixed codebook $C$. For $b$-bit quantization applied to a vector of $d$ contiguous weights, given a codebook $C \in R^{d \times 2^{bd}}$, we have

$$\Phi([w_0, w_1, ..., w_d], C) = C_i. \tag{3}$$

In Equation (3), $C_i$ is the $i$-th codeword in the codebook $C$. For VQ, the quantization process is not performed point-wise. Additionally, the quantization grid is stored within the codebook, which provides a significantly higher degree of freedom. These advantages lead to superior performance at the 2-bit level. Current VQ methods primarily utilize two approaches for quantization grid selection: clustering-based methods and lattice-based methods. Clustering-based methods use algorithms such as K-means to identify representative quantization grids. Lattice-based methods, on the other hand, derive optimal quantization grids specifically for Gaussian distributions.

Regardless of the approach, however, additional storage and more complex decoding processes are inevitable for VQ. A common solution in clustering-based methods is to adopt multi-level codebooks. For example, AQLM (Egiazarian et al., 2024) uses two 1-bit codebooks instead of one 2-bit codebook, reducing the codebook size from $R^{d \times 2^{bd}}$ to $2R^{d \times (2^{b/2 \cdot d})}$. Building upon this, GPTVQ (Baalen et al., 2024) incorporates outlier-aware quantization. Among

lattice-based methods, QuIP# (Tseng et al., 2024a) and NestQuant (Savkin et al., 2025) use codebooks with high spatial symmetry. Meanwhile, Qtip (Tseng et al., 2024b) and CCQ (Zhou et al., 2025) introduce trellis-coded and convolutional codes, respectively. These methods compress codewords and codebooks in various ways, necessitating a decompression step during inference.

## 3. Methods

In this section, we will analyze the differences and connections between scalar and vector quantization. We will also introduce the basic concept of UniSVQ and its details.

### 3.1. The Differences and Connection between Scalar Quantization and Vector Quantization

The differences in quantization grids between VQ and SQ lead to trade-offs in model performance and computational efficiency. In terms of performance, VQ's quantization grids can be positioned in regions of higher probability density, which better leverages the distribution of weight vectors and leads to lower degradation. In terms of efficiency, SQ's quantization grid is derived through simple scaling and translation of integer weights.

In this paper, we propose UniSVQ, a unified representation that combines vector and scalar quantization to leverage the strengths of both. The key insight is selecting a quan-

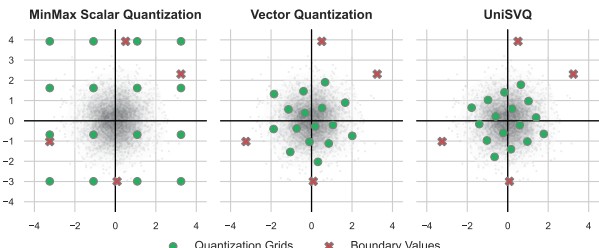

*Figure 2.* Comparison of the 2-dimensional 2-bit quantization grids of scalar quantization, vector quantization, and UniSVQ for an isotropic Gaussian weight. MinMax scalar quantization uses a highly structured grid, which is easier for dequantization but harder to fit the distribution due to boundary values. Vector quantization achieves lower error, but lacks structure. UniSVQ maintains a structured grid while providing higher degrees of freedom.

tization grid that aligns with the weight distribution, while maintaining a structured form that simplifies decoding. We demonstrate that this can be achieved by a linear-constrained quantization grid. Specifically, it is defined as Equation (4):

$$\Phi([w_1, w_2, ..., w_d]^T) = A[\bar{w}_1, \bar{w}_2, ...\bar{w}_d]^T + B. \quad (4)$$

In Equation (4), $\bar{w}_i \in \mathbb{Z}$. This unified representation can be interpreted as either scalar quantization with higher degrees of freedom or vector quantization with constrained codewords. Figure 2 shows the quantization grids of VQ, SQ and UniSVQ. SQ has fixed quantization grids, and the commonly used minmax quantization (Frantar et al., 2022; Xing et al., 2025; Liu et al., 2025a) makes it highly sensitive to the boundary values. Compared to SQ, UniSVQ treats multiple weights as a single entity, replacing the scaling and translation of individual integers with an affine transformation to offer more flexibility. Compared to VQ, UniSVQ has a regular structure, thus reducing storage overhead. For 2-bit quantization with $d = 4$, the required storage can be reduced from $2^{4*2} * 4 * 2 = 2048$ bytes to $(4*4+4)*2 = 40$ bytes, which leads to near 1/64 additional storage reduction.

Furthermore, in UniSVQ, the introduced affine transformation is commutative with the matrix multiplication in linear layers. Therefore, the computational workflow is similar to that of SQ. This allows us to use existing, highly optimized scalar quantization kernels. By imposing specific linear constraints on the quantization grid, we can establish a connection between vector and scalar quantization.

## 3.2. Implementation of UniSVQ

This section presents the details of the UniSVQ method, consisting of three primary stages.

1. **Preprocessing**: We apply a Randomized Hadamard Transform (RHT) to the weights. This step eliminates

outliers and ensures that the linear-constrained quantization grid is sufficiently accurate.

2. **Weight Quantization**: We construct the linear-constrained quantization grid and perform weight quantization using the LDLQ method.

3. **Fine-Tuning**: We perform layer-wise fine-tuning of the affine transformations to minimize reconstruction error further.

### 3.2.1. PREPROCESSING

We first apply a Randomized Hadamard Transform to the weight matrices for preprocessing. Intuitively, outliers can be viewed as vectors with large projections onto a single coordinate axis. The RHT acts as a random rotation of these vectors, redistributing their magnitudes across all axes and eliminating the outliers. The RHT of $W$ is given in Equation (5):

$$R(W) = U S_U W S_V V. \quad (5)$$

In Equation (5), $S_U$ and $S_V$ are diagonal matrices with elements sampled randomly from $\{1, -1\}$ and $U$ and $V$ are Hadamard matrices. Since Hadamard matrices are orthogonal, the RHT is fully reversible. This allows us to perform an inverse transform on the quantized $W_{RHT}$ during inference. Furthermore, computing $U(S_U W)$ using the Fast Walsh–Hadamard Transform (FWHT) takes $O(n log_2 n)$ time. After RHT, $W_{RHT}$ satisfies the incoherence property: $\max(W_{RHT}) \leq \mu ||W||_F / \sqrt{mn}$; Moreover, Tseng et al. (2024a) demonstrate that $W_{RHT}$ approximately follows a standard multi-dimensional Gaussian distribution. Under this condition, the optimal codebook can have regular structure and be approximated by our linear-constrained quantization grid.

### 3.2.2. QUANTIZATION

**Initialization of the linear-constrained quantization grid**. During quantization, UniSVQ maps a group of weights to a vector by passing an integer weight through an affine transformation. The parameters of this transformation determine the structure of the quantization grid, directly impacting the quantization error. First, we must select reasonable initial values for the transformation parameters. This transformation should yield a centrally symmetric quantization grid, and each codeword's values should better follow a Gaussian distribution to match the standard Gaussian distribution of $W_{RHT}$. Specifically, for $b$-bit quantization, the codeword $C_i$ is obtained through the affine transformation:

$$C_i = A\bar{W}_i + B = sG\bar{W}_i - sGb\mathbf{1}. \quad (6)$$

In Equation (6), $\mathbf{1}$ is an all-ones vector, $G$ is a random orthogonal matrix, $b = (2^b - 1)/2$ is a centering constant,

and $s = \sqrt{12/(2^{2b} - 1)}$ is a scaling factor. We prove in Appendix A.1 that this grid satisfies the above requirements.

**Quantization with LDLQ**. After establishing the quantization grid, we perform quantization and calibration using the LDLQ method (Tseng et al., 2024a), which uses the LDL decomposition for the quantization calibration. The reason for choosing LDLQ is its outstanding performance in various VQ methods (Tseng et al., 2024a; Savkin et al., 2025), and UniSVQ is equivalent to VQ using a linear-constrained codebook during quantization. The LDLQ approach performs quantization on weight groups column-by-column, sequentially adjusting the remaining unquantized weights to compensate for the quantization error. After quantizing the $i$-th group of weights, the adjusted weights for the $j$-th column are given by:

$$\hat{W}_j = \Phi\left(W_j + \sum_{k=1}^{j-1}(W_k - \hat{W}_k)a_{kj}\right), \qquad (7)$$

In Equation (7), $j \in \{(i-1)d, \dots, n\}$, $n$ is the total number of columns, and $a_{kj}$ represents the adjustment coefficients. LDLQ computes these adjustment coefficients through the LDL decomposition of the Hessian matrix $H$. Let the Hessian matrix be decomposed as $H = LDL^T$, and the adjustment coefficients $a_j$ correspond to the off-diagonal elements of the decomposition.

### 3.2.3. FINETUNING

The quantization grid is initialized using a random orthogonal matrix, which may not be optimal for minimizing reconstruction error. Although RHT transforms the weights into a nearly isotropic Gaussian distribution, the varying importance of the weights, the distribution of the activations, and the non-flatness inherent to the RHT results (Sun et al., 2025) all influence the optimal configuration of the quantization grid.

To address these issues, we propose a data-driven approach to fine-tune the quantization grid. In UniSVQ, the dequantization process is reformulated as a matrix multiplication, making optimization via backpropagation feasible. Specifically, for a quantized integer matrix $W_{\text{int}}$, the dequantized weight matrix $\hat{W}^T$ is structured as Equation (8):

$$\hat{W}^T = \begin{bmatrix} AW_{\text{int},1}^T + B\mathbf{1}^T \\ AW_{\text{int},2}^T + B\mathbf{1}^T \\ \vdots \\ AW_{\text{int},n/d}^T + B\mathbf{1}^T \end{bmatrix}. \qquad (8)$$

This formulation allows the linear operation $Y = X\hat{W}^T$ to be rewritten as:

$$Y = \sum_{i=1}^{n/d} X_i(AW_{\text{int},i}^T + B\mathbf{1}^T) \\ = \sum_{i=1}^{n/d}(X_iA)W_{\text{int},i}^T + \sum_{i=1}^{n/d}(X_iB)\mathbf{1}^T. \qquad (9)$$

In Equation (9), $X_iA$ is a floating-point activation, and $W_{\text{int},i}$ remains an integer matrix. For $X \in \mathbb{R}^{N \times n}$ and $W_{\text{int}} \in \mathbb{Z}^{m \times n}$, the computational complexity of the dequantization transformation is $O(Nnd) + O(Nn)$. Since $d$ usually takes the value of 4 or 8, the additional computational complexity is much smaller than the $O(Nnm)$ matrix multiplication.

This enables us to optimize A and B using a layer-wise mean squared error (MSE) loss. We fine-tune the floating-point parameters layer by layer. Specifically, after quantizing all matrices within a Transformer block, we fine-tune the block's quantization grid using the activations from the previous quantized layer as input and the output of the original FP16 model as target, given the same inputs. We use MSE loss to minimize reconstruction error, allowing the affine parameters to adaptively compensate for quantization error.

## 4. Experiment Settings

### 4.1. Models and Evaluations

We evaluate the effectiveness of UniSVQ across the Qwen-3 (Yang et al., 2025) and Llama-3 (Llama Team, 2024) model families, covering model scales from 4 billion to 32 billion parameters. We randomly sampled 1,024 sequences from the RedPajama dataset (Weber et al., 2024), each with a length of 2,048. We use these samples to compute $H$ for the LDLQ stage and fine-tune the quantization grids.

We report the PPL on the WikiText 2 (Merity et al., 2016) and C4 (Raffel et al., 2020) datasets and 0-shot accuracy on several downstream benchmarks, including the ARC-Easy, ARC-Challenge (Clark et al., 2018), BoolQ (Clark et al., 2019), HellaSwag (Zellers et al., 2019), PIQA (Bisk et al., 2020), and WinoGrande (Sakaguchi et al., 2021) datasets. 0-shot accuracy is evaluated using the LM-Evaluation-Harness toolkit (Gao et al., 2024).

### 4.2. Quantization Settings

The quantization grid is initialized using random orthogonal matrices generated by the SciPy library [1]. To minimize the additional computational overhead introduced by the affine transformation, the vector quantization dimension $d$ is set to 4, so each linear layer introduces only 20 additional floating-point parameters, including $A \in \mathbb{R}^{4 \times 4}$ and $B \in \mathbb{R}^4$. During

---

[1]https://github.com/scipy/scipy

*Table 1.* PPL and 0-Shot QA accuracy of UniSVQ compared with scalar and vector baselines. With only 20 additional parameters and an $O(n)$ computational overhead, UniSVQ outperforms scalar quantization baselines consistently and achieves comparable results compared to vector quantization using the linear-constrained quantization grid with much less storage and a simpler model structure. AE, AC, BQ, HS, WG, and PQ stand for ARC-Easy, ARC-Challenge, BoolQ, HellaSwag, WinoGrande, and PIQA, respectively.

| MODEL | TYPE | METHOD | WIKI↓ | C4↓ | AC↑ | AE↑ | BQ↑ | HS↑ | PQ↑ | WG↑ | AVG.↑ | PER.↑ |
|---|---|---|---|---|---|---|---|---|---|---|---|---|
| QWEN-3-32B | N/A | FP16 | 7.61 | 12.45 | 60.92 | 83.16 | 86.36 | 82.58 | 82.10 | 72.92 | 78.01 | 1.00 |
| | SCALAR | GPTQ | 1.38E4 | 6.04E3 | 25.34 | 25.46 | 40.58 | 25.19 | 50.59 | 49.40 | 36.09 | 0.46 |
| | | QUIP | 16.72 | 21.74 | 34.89 | 44.57 | 53.14 | 62.24 | 69.20 | 53.35 | 52.90 | 0.68 |
| | | SPINQUANT | 10.90 | 22.01 | 43.34 | 63.89 | 84.40 | 67.40 | 72.91 | 67.96 | 66.65 | 0.85 |
| | | OSTQUANT | 14.79 | 22.46 | 46.84 | 67.55 | 80.80 | 68.37 | 76.50 | 69.69 | 68.29 | 0.88 |
| | VECTOR | QUIP# | 9.04 | 14.13 | 58.10 | 80.00 | 87.89 | 78.46 | 79.70 | 73.63 | 76.30 | 0.98 |
| | | AQLM | 10.56 | 15.13 | 58.27 | 80.85 | 86.94 | 76.21 | 77.80 | 72.13 | 75.37 | 0.97 |
| | UNISVQ | PROPOSED | 9.26 | 14.42 | 58.44 | 80.81 | 87.89 | 77.07 | 78.83 | 73.88 | 76.15 | 0.98 |
| QWEN-3-14B | N/A | FP16 | 8.64 | 13.81 | 60.15 | 82.83 | 89.30 | 78.84 | 79.87 | 72.85 | 77.31 | 1.00 |
| | SCALAR | GPTQ | 1.90E3 | 8.49E2 | 25.94 | 25.76 | 42.02 | 24.76 | 50.27 | 47.20 | 37.29 | 0.48 |
| | | QUIP | 17.34 | 23.19 | 29.86 | 46.42 | 61.22 | 57.47 | 66.87 | 57.14 | 53.16 | 0.69 |
| | | SPINQUANT | 13.75 | 20.35 | 38.99 | 63.97 | 79.42 | 56.28 | 70.40 | 65.51 | 62.43 | 0.81 |
| | | OSTQUANT | 20.96 | 30.52 | 43.26 | 69.61 | 84.74 | 59.17 | 73.72 | 66.93 | 66.24 | 0.86 |
| | VECTOR | QUIP# | 10.76 | 16.43 | 53.67 | 76.89 | 87.95 | 71.91 | 76.99 | 71.67 | 73.18 | 0.95 |
| | | AQLM | 14.80 | 17.51 | 50.60 | 75.29 | 87.34 | 69.17 | 76.17 | 70.80 | 71.56 | 0.93 |
| | UNISVQ | PROPOSED | 11.41 | 16.85 | 51.79 | 78.49 | 86.82 | 69.96 | 76.50 | 71.27 | 72.47 | 0.94 |
| QWEN-3-8B | N/A | FP16 | 9.72 | 15.42 | 56.40 | 80.89 | 86.64 | 74.96 | 77.48 | 68.35 | 74.12 | 1.00 |
| | SCALAR | GPTQ | 4.68E4 | 1.68E4 | 26.79 | 25.67 | 42.87 | 25.84 | 52.50 | 50.04 | 37.29 | 0.50 |
| | | QUIP | 27.61 | 34.42 | 24.91 | 31.69 | 54.89 | 41.41 | 59.90 | 50.12 | 43.82 | 0.59 |
| | | SPINQUANT | 17.82 | 37.28 | 30.46 | 48.40 | 67.06 | 47.94 | 63.17 | 58.64 | 52.61 | 0.71 |
| | | OSTQUANT | 26.08 | 39.71 | 34.64 | 60.02 | 73.21 | 50.29 | 68.50 | 58.17 | 57.47 | 0.78 |
| | VECTOR | QUIP# | 12.37 | 18.45 | 46.50 | 68.43 | 83.12 | 66.62 | 74.32 | 66.30 | 67.55 | 0.91 |
| | | AQLM | 18.26 | 20.73 | 45.22 | 72.31 | 73.73 | 60.99 | 73.07 | 64.33 | 64.94 | 0.88 |
| | UNISVQ | PROPOSED | 14.82 | 19.96 | 45.82 | 72.35 | 85.07 | 63.18 | 74.16 | 67.09 | 67.95 | 0.92 |
| QWEN-3-4B | N/A | FP16 | 10.04 | 16.81 | 58.36 | 81.23 | 84.68 | 69.06 | 75.84 | 68.11 | 72.88 | 1.00 |
| | SCALAR | GPTQ | 1.96E5 | 1.16E5 | 26.62 | 26.05 | 43.30 | 26.12 | 51.52 | 48.30 | 36.99 | 0.51 |
| | | QUIP | 37.88 | 46.59 | 23.98 | 35.82 | 48.29 | 37.38 | 57.67 | 49.41 | 42.09 | 0.58 |
| | | SPINQUANT | 25.88 | 73.35 | 28.92 | 41.05 | 64.34 | 40.74 | 59.19 | 51.78 | 47.67 | 0.65 |
| | | OSTQUANT | 58.85 | 75.13 | 29.35 | 32.78 | 62.17 | 34.66 | 57.62 | 53.43 | 45.00 | 0.62 |
| | VECTOR | QUIP# | 14.78 | 21.25 | 45.82 | 70.41 | 79.72 | 58.55 | 73.01 | 62.75 | 65.04 | 0.89 |
| | | AQLM | 38.16 | 26.99 | 40.96 | 66.88 | 77.22 | 53.02 | 68.61 | 60.85 | 61.26 | 0.84 |
| | UNISVQ | PROPOSED | 20.04 | 23.44 | 43.34 | 65.82 | 82.26 | 55.27 | 70.89 | 62.35 | 63.32 | 0.87 |

quantization, all of the Hessian matrices $H$ are calculated offline, and quantization is performed layerwise to every linear layer in all the transformer blocks. To ensure positive definiteness of $H$ during LDL decomposition, a Tikhonov regularization term $\mu I$ is applied, where $\mu = 0.01$. We fine-tune the linear-constrained quantization grid using the same 1,024 samples partitioned into training and validation sets at a ratio of 7:1. We use a batch size of 16 and a learning rate of $5e - 5$. Each layer is optimized for up to 5 epochs, and the early stop threshold is set to 3. The entire quantization and fine-tuning process takes approximately 6 hours on an Nvidia A100 GPU for an 8B model.

### 4.3. Baselines

Since UniSVQ has a similar structure to orthogonal-transformation-based scalar quantization, we mainly se-

lected SOTA methods in this field as our primary baselines. Furthermore, we include representative VQ methods in our evaluation. Compared to UniSVQ, these VQ methods have more flexible, unconstrained quantization grids, thereby providing an empirical lower bound for degradation.

All methods were evaluated at the 2-bit level. For the scalar quantization baselines, we selected GPTQ (Frantar et al., 2022), Quip (Chee et al., 2023), SpinQuant (Liu et al., 2025a), and OSTQuant (Xing et al., 2025). GPTQ is a classical PTQ strategy that performs quantization and calibration directly on the original weight matrices. QuIP, SpinQuant, and OSTQuant are the orthogonal-transformation-based methods. QuIP uses the RHT, and SpinQuant uses learnable orthogonal transformations. OSTQuant introduces the Quantization Space Utilization Rate (QSUR) metric to optimize the quantization grid. For vector quantization

baselines, we selected AQLM (Egiazarian et al., 2024) and Quip# (Tseng et al., 2024a). AQLM is a representative clustering-based VQ method. We use the standard 2×8 configuration, which uses two 8-bit codewords for an 8-dimensional vector. For QuIP#, we use the E8P codebook proposed by the authors without additional residual quantization. To ensure a fair comparison, all baseline methods use the same calibration data to maintain consistency with our proposed method.

# 5. Results

## 5.1. Main Results

Table 1 compares the PPL and accuracy of the UniSVQ method with that of the baseline methods.

**Comparison with Scalar Quantization**. As the table shows, UniSVQ outperforms all SQ baselines across various model sizes. Under the challenging 2-bit quantization setting, traditional scalar methods experience significant performance degradation. In particular, the unoptimized GPTQ method often yields extremely high PPL and random results. In contrast, UniSVQ maintains high accuracy, retaining up to 98% of the full-precision FP16 model's performance.

Furthermore, UniSVQ achieves consistently superior performance compared to OSTQuant and SpinQuant, which also use orthogonal transformations and fine-tuning, while introducing only 20 additional parameters per weight matrix. This demonstrates the effectiveness of the flexible quantization grid and the strategy of quantizing a group of weights to minimize quantization error. Notably, the 2-bit Qwen-3-32B model quantized by UniSVQ outperforms the FP16 Qwen-3-4B model in terms of average QA accuracy. This suggests that, when GPU memory is limited, deploying a larger 2-bit UniSVQ model is more effective than using a smaller FP16 model. This cannot be achieved by SQ baselines.

**Comparison with Vector Quantization**. Compared to VQ methods, UniSVQ achieves comparable or even superior accuracy while using a linear-constrained quantization grid. Additionally, UniSVQ requires only 1/64 of the codebook storage required by baseline VQ methods. Specifically, UniSVQ yields higher accuracy than the clustering-based AQLM. When compared to QuIP#, which uses a mathematically optimal E8P grid, UniSVQ achieves comparable results, surpassing QuIP# on the Qwen-3-8B model. These results suggest that the performance trade-off introduced by linear constraints is minimal. QuIP# uses a highly flexible E8P lattice codebook that minimizes quantization error without structural constraints, at the cost of complex decoding and higher memory traffic. UniSVQ instead imposes a linear constraint on the codebook, which introduces a modest accuracy trade-off but enables significantly simpler inference. Comparison with other VQ baselines is provided in Appendix A.2.3.

*Table 2.* Ablation study on the fine-tuning and initialization of the quantization grid for Qwen-3-8B. Disabling fine-tuning noticeably degrades zero-shot QA performance. Additionally, replacing the random orthogonal matrix with the D4 lattice generation matrix during initialization results in an even greater performance drop. These results validate the effectiveness of fine-tuning and the rationale behind the proposed initialization strategy.

| METHOD | AVG. | PER. |
|---|---|---|
| PROPOSED | 67.95 | 0.92 |
| W/O FINE-TUNING | 66.99 | 0.90 |
| W/O ORTHOGONAL INIT | 61.00 | 0.82 |

## 5.2. Ablation Studies

### 5.2.1. INFLUENCE OF THE QUANTIZATION GRID

To isolate the contribution of the linear-constrained codebook from other pipeline components, we evaluate each quantizer independently by measuring the weight-level MSE and SNR on Qwen-3-4B at 2-bit. Result in Table 4 shows that UniSVQ achieves substantially lower MSE and an 11.9 dB higher SNR than both GPTQ and SpinQuant, directly demonstrating the superiority of the linear-constrained codebook. This result is consistent with the PPL and zero-shot accuracy improvements reported in Table 1.

We further analyze the influence of fine-tuning the linear-constrained quantization grid and the reasonability of using an orthogonal matrix for initialization. Table 2 shows the accuracy of the 6 0-shot QA tasks for the Qwen-3-8B model.

These results demonstrate that fine-tuning the affine matrix during quantization can increase the 0-shot accuracy. This suggests that optimizing the affine parameters could compensate for performance degradation introduced by the heterogeneity of weight and activation distributions.

To demonstrate the reasonability of using a random orthogonal matrix for initialization, we replace the orthogonal matrix with the generator matrix of the $D4$ lattice, which is proved to be mathematically optimal for 4-dimensional VQ (Tseng et al., 2024a). However, our results show that using the $D4$ lattice generator actually leads to poorer performance. This degradation is likely due to linear constraints. Using the orthogonal matrix to map the integer codebook $\{\bar{W} \mid \bar{W}_i \in \{0, 1, 2, 3\}\}$ results in a diagonal covariance matrix of the codewords, namely $\Sigma_{\hat{W}} = s^2 A \Sigma_{\bar{W}} A^T = s^2 \sigma_{\bar{w}}^2 I$. This preserves the isotropic nature of the grid, making it suitable for quantizing approximately Gaussian and incoherent weights in $W_{\text{RHT}}$. However, when the codewords are generated from the set of integers, the $D4$ generator matrix results in a non-diagonal covariance matrix, which introduces undesirable correlations. These findings confirm that a random orthogonal matrix is a reasonable initial value for a linear-constrained quantization grid compared to the mathematically optimal choice.

*Table 3.* 0-shot QA performance on LLama3-8B-Instruct. UniSVQ outperforms all scalar quantization baselines and is comparable to other vector quantization methods.

| TYPE | METHOD | WIKI↓ | C4↓ | AC↑ | AE↑ | BQ↑ | HS↑ | PQ↑ | WG↑ | AVG.↑ | PER.↑ |
|---|---|---|---|---|---|---|---|---|---|---|---|
| N/A | FP16 | 7.72 | 11.39 | 56.97 | 80.93 | 86.61 | 74.94 | 77.80 | 67.88 | 74.12 | 1.00 |
| SCALAR | GPTQ | 2.55E6 | 6.78E5 | 25.93 | 25.08 | 46.20 | 26.27 | 51.63 | 49.88 | 37.50 | 0.51 |
| | QUIP | 79.63 | 82.83 | 23.54 | 28.66 | 44.55 | 34.71 | 51.57 | 49.17 | 38.70 | 0.52 |
| | SPINQUANT | 27.60 | 96.62 | 21.41 | 33.45 | 60.73 | 39.93 | 56.20 | 55.16 | 44.48 | 0.60 |
| | OSTQUANT | 37.35 | 72.33 | 25.26 | 39.90 | 61.99 | 38.59 | 61.21 | 53.28 | 46.71 | 0.63 |
| VECTOR | QUIP# | 9.42 | 14.28 | 47.44 | 73.27 | 80.21 | 70.70 | 78.13 | 69.53 | 69.72 | 0.94 |
| | AQLM | 27.60 | 96.92 | 32.76 | 53.41 | 78.38 | 63.60 | 67.41 | 63.46 | 59.84 | 0.81 |
| UNISVQ | PROPOSED | 10.70 | 15.43 | 44.79 | 69.52 | 81.77 | 67.20 | 76.27 | 66.14 | 67.62 | 0.91 |

*Table 4.* Quantizer-level isolation on Qwen-3-4B at 2-bit. $H$: Hessian compensation; $R$: Randomized Hadamard Transform; $C$: linear-constrained codebook. "Comp." is the abbreviation of "Components".

| METHOD | COMP. | MSE↓ | SNR (DB)↑ |
|---|---|---|---|
| RTN | — | $9.81 \times 10^{-2}$ | $-12.72$ |
| GPTQ | $H$ | $1.12 \times 10^{-3}$ | $-3.45$ |
| SPINQUANT | $H + R$ | $1.06 \times 10^{-3}$ | $-3.06$ |
| UNISVQ | $H + R + C$ | $7.40 \times 10^{-5}$ | $8.48$ |

*Table 6.* 0-shot QA accuracy for UniSVQ with different $d$ on Qwen-3-8B. Increasing the $d$ provides marginal accuracy improvements, but requires higher computational complexity.

| $d$ | FINE-TUNING | COST | AVG. | PER. |
|---|---|---|---|---|
| 4 | W/O | $O(4Nn)$ | 66.99 | 0.90 |
| 8 | W/O | $O(8Nn)$ | 67.34 | 0.91 |
| 4 | W/ | $O(4Nn)$ | 67.95 | 0.92 |

*Table 5.* Ablation on the Randomized Hadamard Transform on Llama-3-8B at 2-bit. Removing RHT leads to near-random performance.

| METHOD | WIKI↓ | AVG. | PER. |
|---|---|---|---|
| FP16 | 10.04 | 72.88 | 1.00 |
| GPTQ | 19603.88 | 36.99 | 0.51 |
| UNISVQ | 20.04 | 63.42 | 0.87 |
| UNISVQ W/O RHT | 8314.12 | 39.06 | 0.54 |

### 5.2.2. GENERALIZATION ON MODEL ARCHITECTURE

Table 3 presents results on Llama-3-8B-Instruct. UniSVQ achieves a WikiText PPL of 10.70 and an average zero-shot accuracy of 67.62%, outperforming all SQ baselines by a large margin. Specifically, all SQ methods (GPTQ, QuIP, SpinQuant, OSTQuant) suffer severe accuracy degradation at 2-bit, while UniSVQ maintains stable performance and also surpasses AQLM among VQ methods, remaining competitive with QuIP#. These results confirm that UniSVQ generalizes reliably across different Transformer-based model families and training paradigms.

### 5.2.3. INFLUENCE OF VECTOR DIMENSION

In our primary experiments, we set $d = 4$ to minimize the additional parameters and computational overhead introduced by the affine operations. Theoretically, higher dimensions typically yield lower errors (Savkin et al., 2025).

This is further supported by the findings of Tseng et al. (2024a), where the 8-dimensional $E_8$ codebook has better performance over the 4-dimensional $D_4$ codebook.

Table 6 shows the performance difference between 4-dimensional and 8-dimensional UniSVQ. To isolate the direct impact of dimensionality, the results in this table were obtained without fine-tuning. We observe that for Qwen-3-8B, increasing the vector dimension provides only marginal performance gains. Notably, the benefits of higher dimensionality are outweighed by the improvements gained through fine-tuning, while the former has higher additional computational costs. Consequently, we conclude that $d = 4$ represents a better balance between precision and efficiency for the UniSVQ framework.

### 5.2.4. INFLUENCE OF THE RANDOMIZED HADAMARD TRANSFORM

Our codebook design assumes weights after the Randomized Hadamard Transform follow an approximately Gaussian distribution. To evaluate its necessity, we ablate by removing RHT entirely.

As shown in Table 5, without RHT, performance collapses to near-random levels comparable to uncalibrated GPTQ. This confirms that RHT is a necessary prerequisite for the linear-constrained codebook: without it, weight outliers violate the near-Gaussian assumption underlying the lattice initialization and make the affine parameterization ineffective.

*Table 7.* PPL and 0-shot QA accuracy of UniSVQ at different bit-widths on Llama-3-8B.

| BITS | WIKI PPL↓ | AVG. | PER. |
|------|-----------|------|------|
| FP16 | 10.04 | 72.88 | 1.00 |
| 2-BIT | 20.04 | 63.42 | 0.87 |
| 3-BIT | 21.48 | 67.71 | 0.93 |

*Table 8.* Inference throughput and peak GPU memory (GMem) on Llama-3-8B (single A100, batch size 1, 1024 tokens). UniSVQ can achieve 1.68× faster inference and over 75% GMem reduction compared to the FP16 baseline.

| MODEL | THROUGHPUT | PEAK GMEM |
|-------|------------|-----------|
|  | TOK/S (↑) | GB (↓) |
| FP16 | 60.38 | 15.72 |
| GPTQ | 130.70 | 4.14 |
| AQLM 2×8 | 70.97 | 4.44 |
| QUIP# | 79.60 | 4.13 |
| UNISVQ | **101.65** | **3.87** |

### 5.2.5. BIT-WIDTH EXTENSIBILITY

To demonstrate that UniSVQ generalizes beyond the 2-bit setting, we evaluate it under a 3-bit configuration on Llama-3-8B. As shown in Table 7, moving from 2-bit to 3-bit yields consistent gains across all six zero-shot tasks, confirming that UniSVQ effectively utilizes additional bit budget. Furthermore, this scaling requires minimal structural change: the affine parameterization only requires widening the integer vectors, with no modifications to the codebook design. This is a direct advantage over methods such as QuIP# and AQLM, where extending to higher bits requires redesigning the codebook structure.

### 5.2.6. INFERENCE SPEED

Table 8 reports the end-to-end inference throughput and peak GPU memory usage on the Llama-3-8B model. Experiments are conducted on a single NVIDIA A100 GPU, with a batch size of 1 and a generation length of 1024 tokens, averaged over three independent runs. We implement custom fused CUDA kernels for autoregressive generation based on the open-sourced code of Dao-AILab[2].

It can be observed that UniSVQ achieves approximately a 1.68× speedup and over 75% reduction in memory usage compared to the FP16 baseline. Furthermore, UniSVQ maintains a lower memory footprint and higher throughput than AQLM and Quip#. This is possibly attributed to UniSVQ's simple structure and minimal auxiliary parameters. GPTQ achieves a relatively high throughput. However, the severe accuracy degradation of all SQ methods, including GPTQ, makes it less suitable for such extreme low-bit quantization scenarios.

---

[2]https://github.com/Dao-AILab/fast-hadamard-transform

## 6. Conclusion

In this paper, we introduce UniSVQ, a unified representation that bridges scalar and vector quantization. We demonstrate that the key to linking these methods is the linear-constrained quantization grid. With this representation, the dequantization process becomes an affine transformation of integer weights. This allows the model to maintain an architecture nearly as simple as that of SQ and achieve task performance comparable to VQ. Furthermore, we use block-wise fine-tuning to optimize the quantization grid. Experiments on multiple 0-shot QA benchmarks demonstrate that, compared to scalar quantization, UniSVQ introduces only 20 additional parameters per weight matrix yet achieves superior performance. Compared to vector quantization, UniSVQ achieves comparable accuracy while significantly reducing the number of auxiliary parameters.

## Limitations

Our current study is limited to weight-only quantization and does not consider activation or KV-cache quantization, which are also important components for end-to-end inference efficiency in LLMs.

Moreover, while the linear-constrained quantization grid brings modest overhead, their interaction with highly optimized GEMM kernels remains to be explored.

## Acknowledgements

This work is supported by the National Key Research and Development Program of China (2024YFB4505603) and the National Natural Science Foundation of China (No. 62576186). This work is also supported by Tsinghua KA Excellence Center. This work is partially supported by Tsinghua University (Department of Computer Science and Technology) - Sinopec Joint Research Center for Artificial Intelligence.

## Impact Statement

This paper presents work whose goal is to advance the field of Machine Learning. There are many potential societal consequences of our work, none which we feel must be specifically highlighted here.

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

# A. Appendix

## A.1. The initialization of $C_i$

We provide a proof of the mean, covariance and the approximate Gaussianity of the elements in $C_i$, which ensures the resulting quantization grid is a reasonable initialization for a standard Gaussian distributed weight $W_{\text{RHT}}$.

### A.1.1. MEAN OF $C_i$

Assuming $\bar{W}_i^j$ is uniformly distributed over $\{0, 1, \ldots, 2^b - 1\}$, its expectation is $E[\bar{W}_i^j] = (2^b - 1)/2 = b$. Thus, $E[\bar{W}_i] = b\mathbf{1}$. As a result, we have:

$$E[C_i] = sG(E[\bar{W}_i]) - sGb\mathbf{1} = sGb\mathbf{1} - sGb\mathbf{1} = 0$$

The grid is thus centered at zero.

### A.1.2. COVARIANCE OF $C_i$

The variance of a discrete uniform distribution $\bar{W}_i^j$ is $\text{Var}(\bar{W}_i^j) = (2^{2b} - 1)/12$. Let $\sigma_{\bar{W}}^2$ denote this variance; then the covariance matrix of $\bar{W}_i$ is $\Sigma_{\bar{W}} = \sigma_{\bar{W}}^2 I$. The covariance of $C_i$ is:

$$\Sigma_{C_i} = \text{Cov}(sG\bar{W}_i) = s^2 G\Sigma_{\bar{W}}G^T = s^2\sigma_{\bar{W}}^2 GG^T$$

Since $G$ is an orthogonal matrix, $GG^T = I$. As a result, we have

$$s = 1/\sigma_{\bar{W}} = \sqrt{\frac{12}{2^{2b} - 1}}$$

Thus $\Sigma_{C_i} = I$.

### A.1.3. APPROXIMATE GAUSSIANITY OF $C_i$

According to the results regarding the Haar measure on orthogonal groups, the entries of a large $n \times n$ random orthogonal matrix $\mathbf{G}$ are approximately Gaussian with mean 0 and variance $1/n$. Let $\mathbf{g} = \mathbf{G}\bar{\mathbf{W}}_i$ be the product of the matrix and a fixed integer vector. The $k$-th element of $\mathbf{g}$ is given by the inner product:

$$g_k = \sum_{j=1}^{n} G_{kj}\bar{W}_{ij}$$

Since each $G_{kj}$ is approximately Gaussian and the sum represents a linear combination of these components, the resulting element $g_k$ retains Gaussian properties.

## A.2. Detailed Results of the Ablation Studies

In this section, we will present the detailed results of the ablation experiments on the selected 6 evaluation set regarding fine-tuning, orthogonal initialization, and vector dimension, as described in Section 5.2.

### A.2.1. THE EFFECTIVENESS OF FINE-TUNING

Table 9 presents the detailed accuracy of the selected 0-shot QA dataset. The results show that block-wise fine-tuning of the quantization grid improves performance consistently across all tasks, confirming the effectiveness of our optimization strategy. Furthermore, in Section 5.2.1, we replace the random orthogonal initialization with the generation matrix of $D_4$, which is defined as below:

$$A = \begin{bmatrix} 1 & -1 & 0 & 0 \\ 0 & 1 & -1 & 0 \\ 0 & 0 & 1 & -1 \\ 0 & 0 & 1 & 1 \end{bmatrix}$$

This results in consistent performance degradation, particularly on more challenging benchmarks such as ARC-Challenge. These findings further validate that the random orthogonal matrix provides a robust initialization for UniSVQ.

*Table 9.* The detailed 0-shot QA accuracy of the baselines and UniSVQ is shown for variants without fine-tuning or orthogonal initialization. Removing these two steps will result in consistent performance degradation.

| QUANTIZATION TYPE | METHOD | AC | AE | BQ | HS | PQ | WG | AVG. | PER. |
|---|---|---|---|---|---|---|---|---|---|
| N/A | FP16 | 56.40 | 80.89 | 86.64 | 74.96 | 77.48 | 68.35 | 74.12 | 1.00 |
| SCALAR | GPTQ | 26.79 | 25.67 | 42.87 | 25.84 | 52.50 | 50.04 | 37.29 | 0.50 |
| | QUIP | 24.91 | 31.69 | 54.89 | 41.41 | 59.90 | 50.12 | 43.82 | 0.59 |
| | SPINQUANT | 30.46 | 48.40 | 67.06 | 47.94 | 63.17 | 58.64 | 52.61 | 0.71 |
| | OSTQUANT | 34.64 | 60.02 | 73.21 | 50.29 | 68.50 | 58.17 | 57.47 | 0.78 |
| VECTOR | QUIP# | 46.50 | 68.43 | 83.12 | 66.62 | 74.32 | 66.30 | 67.55 | 0.91 |
| | AQLM 2*8 | 45.22 | 72.31 | 73.73 | 60.99 | 73.07 | 64.33 | 64.94 | 0.88 |
| UNISVQ | PROPOSED | 45.82 | 72.35 | 85.07 | 63.18 | 74.16 | 67.09 | 67.95 | 0.92 |
| | W/O TUNING | 45.56 | 69.36 | 83.85 | 62.94 | 73.94 | 66.30 | 66.99 | 0.90 |
| | W/O ORTHOGONAL INIT | 36.52 | 67.55 | 74.19 | 56.05 | 71.12 | 60.54 | 61.00 | 0.82 |

### A.2.2. THE INFLUENCE OF VECTOR DIMENSION $d$

Table 10 shows the detailed accuracy of the selected 0-shot QA dataset with different vector dimension. Using larger $d$ results in some improvement in the average accuracy, but the effect is not consistent, and is weaker than that of Fine-tuning. Considering that the additional computational complexity is $O(Nnd)$, this further indicates that choosing $d = 4$ is a more reasonable option.

*Table 10.* The detailed 0-shot QA accuracy of UniSVQ with different quantization dimensions $d$. Using a larger $d$ results in inconsistent improvement.

| $d$ | FINE-TUNING | AC | AE | BQ | HS | PQ | WG | AVG. | PER. |
|---|---|---|---|---|---|---|---|---|---|
| 4 | W/O | 45.56 | 69.36 | 83.85 | 62.94 | 73.94 | 66.30 | 66.99 | 0.90 |
| 8 | W/O | 46.07 | 72.94 | 80.86 | 62.65 | 73.07 | 68.43 | 67.34 | 0.91 |
| 4 | W | 45.82 | 72.35 | 85.07 | 63.18 | 74.16 | 67.09 | 67.95 | 0.92 |

### A.2.3. COMPARISON WITH SOTA VQ BASELINES

UniSVQ's goal is not to maximize quantization precision at the cost of complexity. Instead, it targets a practical trade-off: the affine parameterization preserves compatibility with scalar quantization matmul kernels, avoiding the decompression overhead typical of advanced VQ methods.

To further situate UniSVQ among more advanced VQ baselines, we include Qtip (Tseng et al., 2024b) as an additional comparison on Qwen-3-8B at 2-bit quantization. As shown in Table 11, Qtip achieves the highest accuracy among all VQ methods, likely due to its trellis-coded codebook, while UniSVQ remains competitive and retains kernel-level compatibility.

*Table 11.* Comparison with SOTA VQ methods on Qwen-3-8B at 2-bit. Per. denotes performance relative to FP16.

| QUANTIZATION TYPE | METHOD | WIKI↓ | C4↓ | AC | AE | BQ | HS | PQ | WG | AVG. | PER. |
|---|---|---|---|---|---|---|---|---|---|---|---|
| N/A | FP16 | 9.72 | 15.42 | 56.40 | 80.89 | 86.64 | 74.96 | 77.48 | 68.35 | 74.12 | 1.00 |
| SCALAR | GPTQ | 46810.97 | 16797.36 | 26.79 | 25.67 | 42.87 | 25.84 | 52.50 | 50.04 | 37.29 | 0.50 |
| | QUIP | 27.61 | 34.42 | 24.91 | 31.69 | 54.89 | 41.41 | 59.90 | 50.12 | 43.82 | 0.59 |
| | SPINQUANT | 17.82 | 37.28 | 30.46 | 48.40 | 67.06 | 47.94 | 63.17 | 58.64 | 52.61 | 0.71 |
| | OSTQUANT | 26.08 | 39.71 | 34.64 | 60.02 | 73.21 | 50.29 | 68.50 | 58.17 | 57.47 | 0.78 |
| VECTOR | QUIP# | 12.37 | 18.45 | 46.50 | 68.43 | 83.12 | 66.62 | 74.32 | 66.30 | 67.55 | 0.91 |
| | AQLM 2×8 | 18.26 | 20.73 | 45.22 | 72.31 | 73.73 | 60.99 | 73.07 | 64.33 | 64.94 | 0.88 |
| | QTIP | 11.55 | 17.59 | 52.05 | 77.06 | 84.95 | 68.14 | 77.04 | 67.25 | 71.08 | 0.96 |
| UNISVQ | PROPOSED | 14.82 | 19.96 | 45.82 | 72.35 | 85.07 | 63.18 | 74.16 | 67.09 | 67.95 | 0.92 |

