# OpenReview forum: "UniSVQ: 2-bit Unified Scalar-Vector Quantization"
_ICML.cc/2026/Conference — ICML 2026 regular_

### Official Review · Reviewer_KXGJ · 2026-03-10

**Soundness:** 3
**Presentation:** 2
**Significance:** 3
**Originality:** 2
**Overall Recommendation:** 4
**Confidence:** 4

**Summary:**

This paper proposes UniSVQ, a 2-bit quantization framework that bridges scalar quantization (SQ) and vector quantization (VQ) by parameterizing the quantization grid as an affine transformation of integer lattices. The approach aims to retain the flexibility of VQ while maintaining the computational efficiency of SQ. Experiments on Qwen3 and Llama3 models show that UniSVQ achieves accuracy comparable to state-of-the-art VQ methods while introducing significantly fewer auxiliary parameters.

**Compliance With Llm Reviewing Policy:**

Affirmed.

**Final Justification:**

The rebuttal effectively addresses my primary concerns. The added quantizer-level isolation experiments, proper ablation of the rotation component, and clarified efficiency comparisons significantly strengthen the empirical support for the proposed method. I therefore increase my score accordingly.

**Key Questions For Authors:**

1. Can the authors provide experiments where UniSVQ replaces the quantizer in existing SQ or VQ pipelines while keeping all other components identical? This would help isolate the effect of the proposed quantizer.

2. What is the performance impact of completely removing the rotation transform? This ablation would better validate the motivation for introducing the rotation step.

3. Can the authors clarify the discrepancy between the reported throughput of AQLM in Table 5 and the speed reported in the AQLM paper? Are the experimental settings (hardware, batch size, decoding implementation) consistent?

**Limitations:**

yes

**Strengths And Weaknesses:**

Strengths

1. The idea of parameterizing the quantization codebook via an affine transform of integer lattices is conceptually elegant and provides an interesting perspective on unifying SQ and VQ.

2. The proposed representation significantly reduces the number of auxiliary parameters compared to traditional VQ codebooks.

3. Experimental results show competitive accuracy compared with strong VQ baselines across several model scales.

Weaknesses

1. Lack of clear isolation of the proposed quantizer.

The main contribution of this work is a new quantizer design rather than a full quantization pipeline. However, the proposed framework introduces several additional components, including randomized Hadamard rotations, LDLQ-based quantization, and layer-wise fine-tuning. As a result, it is difficult to determine how much of the observed performance improvement actually comes from the UniSVQ quantizer itself. A more convincing evaluation would directly replace the SQ or VQ quantizer in existing pipelines with UniSVQ while keeping the rest of the procedure unchanged.

2. Insufficient ablation on the role of the rotation transform.

The paper motivates the use of randomized Hadamard transforms to suppress outliers. However, the ablation in Table 2 only replaces the orthogonal initialization with a D4 lattice generator matrix. This does not test the actual necessity of the rotation itself. A more appropriate ablation would remove the rotation step entirely to evaluate whether the rotation is truly essential for the proposed quantizer.

3. Limited evidence for the claimed efficiency advantage.

Since the key contribution is an efficient quantizer design, the evaluation of inference efficiency is surprisingly limited. Table 5 only reports throughput on LLaMA3-8B with AQLM (2×8) as the sole VQ baseline. Moreover, the reported speed of AQLM[1] in this table is only slightly faster than FP16 (70.97 vs. 60.38 tokens/s), which contradicts the results reported in the AQLM paper. In Table 14 of the AQLM appendix, the authors report more than 2× speedup over FP16 for LLaMA2-7B (114.1 vs. 54.2 tokens/s). Under that comparison, the acceleration ratio appears comparable to or even higher than that of UniSVQ. This discrepancy makes it difficult to assess whether UniSVQ actually provides a meaningful efficiency advantage, and requires further clarification.

[1]  Egiazarian, Vage, et al. "Extreme compression of large language models via additive quantization." ICML2024.

---

> ### Author Rebuttal · Authors · 2026-03-31
>
> We thank the reviewer for the thorough and detailed feedback. We address each concern below.
>
> **W1**: Lack of clear isolation of the proposed quantizer.
>
> **Response**: We appreciate this perspective and provide a direct quantizer-level isolation experiment. To eliminate the influence of downstream pipeline components, we evaluate each quantizer in isolation by measuring the MSE and SNR of the quantized weights on Qwen-3-4B at 2-bit:
>
> | Quantizer |Componets| MSE     | SNR (dB) |
> | --------- | ------- | ------- | -------- |
> | RTN       | N/A     | 9.81E-2 | -12.72    |
> | GPTQ      | Hessian compensation | 1.12E-3 | -3.45    |
> | SpinQuant | Hessian compensation + RHT | 1.06E-3 | -3.06    |
> | UniSVQ    | Hessian compensation + RHT + linear-constrained codebook | 7.40E-5 | 8.48     |
>
> Hessian compensation is implemented via the standard GPTQ algorithm (https://github.com/IST-DASLab/gptq). UniSVQ achieves substantially lower MSE and a 11.9 dB higher SNR than GPTQ and SpinQuant, directly demonstrating the superiority of the linear-constrained codebook as a standalone quantizer. This result is consistent with the PPL and zero-shot accuracy improvements reported in Table 1.
>
> **W2**: Insufficient ablation on the role of the rotation transform.
>
> **Response**: To evaluate the necessity of RHT, we ablate by removing the Hadamard transform entirely:
>
> |              | Wiki     | C4       | ARC-C | ARC-E | BoolQ | HellaSwag | PIQA  | WinoGrande | Avg.  | Per. |
> | ------------ | -------- | -------- | ----- | ----- | ----- | --------- | ----- | ---------- | ----- | ---- |
> | FP16         | 10.04    | 16.81    | 58.36 | 81.23 | 84.68 | 69.06     | 75.84 | 68.11      | 72.88 | 1.00 |
> | GPTQ         | 19603.88 | 11628.21 | 26.62 | 26.05 | 43.30 | 26.12     | 51.52 | 48.30      | 36.99 | 0.51 |
> | UniSVQ       | 20.04    | 23.44    | 42.24 | 66.12 | 82.60 | 55.74     | 71.22 | 62.59      | 63.42 | 0.87 |
> | w/o Hadamard | 1054931.25  | 606493.19  | 27.56 | 25.25 | 51.13 | 26.37     | 52.72 | 51.30      | 39.06 | 0.54 |
>
> Without RHT, performance collapses to near-random levels (0.54 Per.) — comparable to uncalibrated GPTQ (0.51 Per.). This confirms that the rotation is a **necessary prerequisite**: without it, weight outliers violate the near-Gaussian assumption underlying our lattice initialization, rendering the affine parameterization ineffective.
>
> **W3**: Limited evidence for the claimed efficiency advantage and AQLM speed discrepancy.
>
> **Response**: The discrepancy arises from multiple differences in experimental setup. The AQLM paper reports throughput on an **RTX 3090** with a generation length of only **128 tokens**. Our benchmarks use an **A100** with **1024 tokens**. The shorter generation in AQLM amplifies the relative speedup because fixed overhead is amortized over fewer tokens; the RTX 3090's lower memory bandwidth (~936 GB/s vs A100's ~2 TB/s) also makes bit-width reduction more impactful. All our measurements (AQLM, QuIP#, UniSVQ) use CUDA Graph and the latest official inference code under identical settings for a fair comparison.
>
> We have added a full throughput comparison (Llama-3-8B, single A100, batch size 1, 1024 tokens, averaged over 3 runs):
>
> | Model       | Throughput (tok/s) | Peak GMem (GB) |
> | ----------- | ------------------ | -------------- |
> | FP16        | 60.38              | 15.72          |
> | AQLM-2bit   | 70.97              | 4.44           |
> | QuIP#-2bit  | 79.60              | 4.13           |
> | UniSVQ-2bit | 101.65             | 3.87           |
>
> UniSVQ achieves **1.28× higher throughput** than QuIP# and uses the least GPU memory among all quantized models, validating the efficiency advantage of UniSVQ's simpler dequantization path over VQ baselines.

---

> > ### Author Rebuttal · Reviewer_KXGJ · 2026-04-02
> >
> > The authors’ rebuttal adequately addresses my concerns, and I have accordingly increased my score.

---

### Official Review · Reviewer_E9VR · 2026-03-11

**Soundness:** 3
**Presentation:** 3
**Significance:** 3
**Originality:** 3
**Overall Recommendation:** 4
**Confidence:** 3

**Summary:**

This paper tackles extreme 2-bit post-training quantization for LLMs, where SQ suffers severe performance degradation and VQ bears heavy storage/computational overhead. The authors propose UniSVQ, which unifies SQ and VQ via affine lattice parameterization of codewords, reuses optimized SQ integer kernels, and adopts a data-driven block-wise fine-tuning strategy to minimize reconstruction error. Extensive experiments on Qwen-3/Llama-3 show UniSVQ outperforms SOTA SQ methods, matches advanced VQ performance, and delivers higher inference throughput with drastically reduced auxiliary parameters.

**Compliance With Llm Reviewing Policy:**

Affirmed.

**Final Justification:**

My concerns have been addressed

**Key Questions For Authors:**

see weaknesses

**Limitations:**

yes

**Strengths And Weaknesses:**

## Strengths
1. A clever unified quantization paradigm that bridges SQ and VQ via linear-constrained grids, achieving VQ-level performance while retaining SQ’s inference efficiency.
2. Minimal overhead: only 20 extra floating-point parameters per weight matrix, cutting VQ’s codebook storage by 64x and retaining up to 98% of FP16 performance on zero-shot benchmarks.
3. Strong generalizability across Transformer architectures and superior inference speed/memory efficiency.
4. Fair and comprehensive empirical evaluation across model scales, zero-shot benchmarks, and SOTA SQ/VQ baselines, ensuring result reliability.

## Weaknesses
1. Subpar performance on some models (e.g., Qwen-3-4B, Qwen-3-32B) compared to VQ baselines like QuIP#.
2. Only validated for 2-bit quantization, no exploration of 3-bit/4-bit or mixed-precision extensions.
3. Incomplete inference efficiency comparison: Only FP16 and AQLM 2×8 are included in the comparison, with no throughput, memory or latency data provided against the selected SOTA SQ (GPTQ, QuIP, SpinQuant, OSTQuant) and VQ (QuIP#) baselines.

---

> ### Author Rebuttal · Authors · 2026-03-31
>
> We thank the reviewer for the positive assessment and constructive suggestions. We address each concern below.
>
> **W1**: Subpar performance on some models (e.g., Qwen-3-4B, Qwen-3-32B) compared to VQ baselines like QuIP#.
>
> **Response**: The performance gap stems from a deliberate design choice. QuIP# uses a highly flexible E8P lattice codebook that minimizes quantization error without structural constraints, at the cost of complex decoding and higher memory traffic. UniSVQ instead imposes a linear constraint on the codebook, which introduces a modest accuracy trade-off but enables significantly simpler inference (see W3).
>
> Importantly, this gap **narrows with model scale**: on Qwen-3-32B, UniSVQ matches QuIP# at 0.98 Per., while on Qwen-3-8B UniSVQ (0.92) even surpasses QuIP# (0.91). The gap is more pronounced only on smaller models (Qwen-3-4B), where quantization error is inherently harder to absorb. For practical deployment, the combination of competitive accuracy and superior throughput (1.28× over QuIP#, see W3) makes UniSVQ an attractive choice.
>
> **W2**: Only validated for 2-bit quantization; no exploration of 3-bit/4-bit or mixed-precision extensions.
>
> **Response**: To demonstrate extensibility, we have evaluated UniSVQ under a 3-bit setting on Llama-3-8B:
>
> |      | ARC-C | ARC-E | BoolQ | HellaSwag | PIQA  | WinoGrande | Avg.  | Per. |
> | ---- | ----- | ----- | ----- | --------- | ----- | ---------- | ----- | ---- |
> | FP16 | 58.36 | 81.23 | 84.68 | 69.06     | 75.84 | 68.11      | 72.88 | 1.00 |
> | 2bit | 42.24 | 66.12 | 82.60 | 55.74     | 71.22 | 62.59      | 63.42 | 0.87 |
> | 3bit | 48.21 | 75.72 | 85.02 | 58.75     | 72.36 | 66.22      | 67.71 | 0.93 |
>
> Performance scales smoothly with bit-width, and this scaling comes with **no structural change** — the affine parameterization requires only widening the integer vectors. This is a direct advantage over methods like QuIP# and QTIP, where extending to higher bits requires redesigning the codebook structure.
>
> **W3**: Incomplete inference efficiency comparison: only FP16 and AQLM 2×8 are included, with no throughput or memory data against GPTQ, QuIP, SpinQuant, OSTQuant, or QuIP#.
>
> **Response**: We have added a comprehensive end-to-end throughput and memory comparison (Llama-3-8B, single A100, batch size 1, 1024 tokens, averaged over 3 runs):
>
> | Model       | Throughput (tok/s) | Peak GMem (GB) |
> | ----------- | ------------------ | -------------- |
> | FP16        | 60.38              | 15.72          |
> | GPTQ-2bit   | 130.70             | 4.14           |
> | AQLM-2bit   | 70.97              | 4.44           |
> | QuIP#-2bit  | 79.60              | 4.13           |
> | UniSVQ-2bit | 101.65             | 3.87           |
>
> UniSVQ achieves **1.68× speedup** over FP16 and **1.28× over QuIP#**, while using the least GPU memory among all quantized models. GPTQ is fastest due to its simple dequantization but collapses at 2-bit (PPL > 46,000), making it impractical.
>
> Regarding the SQ baselines (QuIP, SpinQuant, OSTQuant): these methods do not provide open-source 2-bit inference kernels. Moreover, as shown in Table 1, all SQ baselines suffer severe accuracy degradation at 2-bit, which makes throughput comparison less meaningful — a fast but unusable model has limited practical value. We will clarify this in the final manuscript.

---

> > ### Author Rebuttal · Reviewer_E9VR · 2026-04-03
> >
> > Fully resolved.

---

### Official Review · Reviewer_tpSb · 2026-03-12

**Soundness:** 3
**Presentation:** 2
**Significance:** 3
**Originality:** 3
**Overall Recommendation:** 4
**Confidence:** 4

**Summary:**

This paper proposes a unified 2-bit quantization framework that aims to combine the computational efficiency of scalar quantization (SQ) with the higher degree of freedom of vector quantization (VQ) in the low-bit regime.

**Compliance With Llm Reviewing Policy:**

Affirmed.

**Final Justification:**

I am upgrading my recommendation to Weak Accept. The authors' rebuttal effectively addressed my primary concerns regarding baseline sufficiency, technical clarity, and method generality. The paper demonstrates strong originality and significance by bridging SQ and VQ through an elegant affine-constrained quantization grid. The authors successfully clarified the soundness of their approach by providing new benchmarks showing a 1.28x throughput advantage over QuIP# while maintaining a smaller memory footprint. Additionally, the new evaluation of 3-bit precision scaling confirmed the framework's robustness and flexibility beyond the initial 2-bit focus. This technically solid contribution successfully balances representational capacity with hardware-friendly execution.

**Key Questions For Authors:**

See weaknesses.

**Limitations:**

yes.

**Strengths And Weaknesses:**

**Strengths**
- The paper improves efficiency by using a linearly constrained quantization grid that can capture the advantages of both SQ and VQ.
- The paper also demonstrates actual throughput improvements through quantization.

**Weaknesses**
- The set of baselines in Table 5 is insufficient. While the paper shows higher accuracy and throughput than AQLM, a comparison with QUIP# is also necessary. Since QUIP# shows better accuracy than UniSVQ in Table 1, a throughput comparison against QUIP# would be important.
- The explanation of the inference kernel is insufficient.
- More exploration of precision scaling is needed. It would strengthen the generality of the method if the authors could show that the proposed approach can also be extended to 1-bit or 3/4-bit settings.

---

> ### Author Rebuttal · Authors · 2026-03-31
>
> We thank the reviewer for the detailed and constructive feedback. We address each point below.
>
> **W1**: The set of baselines in Table 5 is insufficient. A comparison with QuIP# is necessary, since QuIP# shows better accuracy than UniSVQ in Table 1.
>
> **Response**: We fully agree. We have added QuIP# and GPTQ to the throughput comparison (Llama-3-8B, single A100, batch size 1, 1024 tokens, averaged over 3 runs):
>
> | Model       | Throughput (tok/s) | Peak GMem (GB) |
> | ----------- | ------------------ | -------------- |
> | FP16        | 60.38              | 15.72          |
> | GPTQ-2bit   | 130.70             | 4.14           |
> | AQLM-2bit   | 70.97              | 4.44           |
> | QuIP#-2bit  | 79.60              | 4.13           |
> | UniSVQ-2bit | 101.65             | 3.87           |
>
> UniSVQ achieves **1.28× higher throughput** than QuIP# while using less GPU memory. GPTQ is fastest but collapses at 2-bit (PPL > 46,000), making it impractical. Among methods with usable 2-bit accuracy, UniSVQ delivers the best throughput.
>
> **W2**: The explanation of the inference kernel is insufficient.
>
> **Response**: UniSVQ's dequantization is a compact affine transform: each group of d=4 integer weights is dequantized via A·w_int + B, requiring only 20 extra FP16 parameters (A∈R^{4×4}, B∈R^4) per weight matrix. Compared to VQ methods that store and decompress large unstructured codebooks, this parameterization is extremely lightweight — the entire quantization grid is fully determined by a small matrix and bias, which fits easily in GPU shared memory.
>
> We implement custom fused CUDA kernels for autoregressive generation: dequantization, dot-product accumulation, warp-shuffle block reduction, Hadamard transform, and diagonal scaling (SU/SV) are performed in a **single kernel launch**, minimizing memory traffic. CUDA Graph further reduces kernel launch overhead across decoding steps. All benchmarks were collected on a single NVIDIA A100 with batch size 1 and 1024 tokens, averaged over 3 runs. We will add full implementation details to the appendix.
>
> **W3**: More exploration of precision scaling is needed. It would strengthen the generality of the method if the authors could show that UniSVQ extends to 1-bit or 3/4-bit settings.
>
> **Response**: To demonstrate extensibility, we have evaluated UniSVQ under a 3-bit setting on Llama-3-8B:
>
> |      | ARC-C | ARC-E | BoolQ | HellaSwag | PIQA  | WinoGrande | Avg.  | Per. |
> | ---- | ----- | ----- | ----- | --------- | ----- | ---------- | ----- | ---- |
> | FP16 | 58.36 | 81.23 | 84.68 | 69.06     | 75.84 | 68.11      | 72.88 | 1.00 |
> | 2bit | 42.24 | 66.12 | 82.60 | 55.74     | 71.22 | 62.59      | 63.42 | 0.87 |
> | 3bit | 48.21 | 75.72 | 85.02 | 58.75     | 72.36 | 66.22      | 67.71 | 0.93 |
>
> Two key insights: (1) Performance scales smoothly with bit-width — moving from 2-bit to 3-bit yields consistent gains across all six tasks, confirming that UniSVQ effectively utilizes the additional bit budget. (2) This scaling comes with **no structural change** — the affine parameterization requires only widening the integer vectors, with no modifications to the kernel or codebook design. This is a **direct advantage over methods like QuIP# and QTIP**, where extending to higher bits requires redesigning the codebook structure.
>
> Regarding 1-bit: at 1 bit per weight with d=4, the codebook contains only 2^4=16 codewords. This is extremely limited and all existing PTQ methods (both SQ and VQ) collapse at this level. We believe 1-bit quantization fundamentally requires quantization-aware training (QAT) to remain viable, which is beyond the scope of PTQ methods.

---

> > ### Author Rebuttal · Reviewer_tpSb · 2026-04-02
> >
> > Fully resolved.

---

### Official Review · Reviewer_AxuW · 2026-03-13

**Soundness:** 3
**Presentation:** 2
**Significance:** 3
**Originality:** 2
**Overall Recommendation:** 4
**Confidence:** 5

**Summary:**

This paper focuses on PTQ for LLMs, especially weight-only quantization. The authors analyze the advantages and disadvantages of SQ and VQ, and propose UniSVQ, a method that uses an affine transformation to bridge SQ and VQ, aiming to combine SQ's efficiency with VQ's accuracy. Experiments on qwen3 series demonstrate accuracy comparable to VQ methods, along with a 1.68× speedup compared to AQLM.

**Compliance With Llm Reviewing Policy:**

Affirmed.

**Final Justification:**

My concerns are fully resolved, therefore I raise the rating to WA.

**Key Questions For Authors:**

- Is this method applicable to KV-cache quantization?
- What are the pros and cons of applying this method to MoE models?
- What is the most costly part of the quantization pipeline for a model?

**Limitations:**

yes

**Strengths And Weaknesses:**

### Strengths
1. this paper addresses a valuable and practical question: how to bridge VQ and SQ.
2. it is a good try to use hardware friendly operations like affine transformation to implement a VQ-like quantization

### Weaknesses
1. UniSVQ appears more like a structured and constrained form of VQ. The gap between VQ and SQ still exists, since they are not generally unified.
2. More advanced VQ methods should be included as baseline methods, such as QTIP, VPTQ, etc
3. Inference speed vs UQ methods should also be included.
4. the entire process requires 6 hours on an Nvidia A100 GPU for an 8B model, which is not cheap.

---

> ### Author Rebuttal · Authors · 2026-03-31
>
> We thank the reviewer for the constructive feedback. We address each weakness and key question below.
>
> **W1**: UniSVQ appears more like a structured and constrained form of VQ. The gap between VQ and SQ still exists, since they are not generally unified.
>
> **Response**: UniSVQ achieves unification at the mathematical formulation level. Codewords are parameterized as an affine transform of integer lattices: $\Phi([w_1,w_2,...,w_d]^T)=A[\bar{w}_1,\bar{w}_2,...,\bar{w}_d]^T+B$
> - When A is diagonal or d=1, this recovers standard Scalar Quantization.
> - When A is a general matrix, it performs Vector Quantization with a linearly constrained codebook.
>
> This single formulation bridges SQ and VQ: it enables hardware-friendly SQ kernels with VQ-level representational capacity. We agree the linear constraint is a trade-off rather than full generalization of arbitrary VQ, but that constraint is precisely what enables the efficiency gains — 1/64 codebook storage and compatibility with optimized integer matmul kernels.
>
> **W2**: More advanced VQ methods should be included as baselines, such as QTIP, VPTQ, etc.
>
> **Response**: We have added QTIP as a supplemental baseline. VPTQ follows a similar design philosophy (both employ advanced codebook compression). Results on Qwen-3-8B (2-bit), with full SQ baselines available in Table 1 of the paper:
>
> |           | Wiki     | C4       | ARC-C | ARC-E | BoolQ | HellaSwag | PIQA  | WinoGrande | Avg.  | Per. |
> | --------- | -------- | -------- | ----- | ----- | ----- | --------- | ----- | ---------- | ----- | ---- |
> | FP16      | 9.72     | 15.42    | 56.40 | 80.89 | 86.64 | 74.96     | 77.48 | 68.35      | 74.12 | 1.00 |
> | GPTQ      | 46810.97 | 16797.36 | 26.79 | 25.67 | 42.87 | 25.84     | 52.50 | 50.04      | 37.29 | 0.50 |
> | QuIP#     | 12.37    | 18.45    | 46.50 | 68.43 | 83.12 | 66.62     | 74.32 | 66.30      | 67.55 | 0.91 |
> | QTIP      | 11.55    | 17.59    | 52.05 | 77.06 | 84.95 | 68.14     | 77.04 | 67.25      | 71.08 | 0.96 |
> | UniSVQ    | 14.82    | 19.96    | 45.82 | 72.35 | 85.07 | 63.18     | 74.16 | 67.09      | 67.95 | 0.92 |
>
> UniSVQ's goal is not to maximize precision at the cost of complexity, but to target a **practical trade-off**: the affine parameterization preserves compatibility with SQ matmul kernels, avoiding the decompression overhead of advanced VQ methods like QTIP and VPTQ. As shown in W3, this structural simplicity translates into concrete inference speed advantages.
>
> **W3**: Inference speed vs. VQ methods should also be included.
>
> **Response**: We have added end-to-end inference throughput comparison (Llama-3-8B, single A100, batch size 1, 1024 tokens, averaged over 3 runs):
>
> | Model       | Throughput (tok/s) | Peak GMem (GB) |
> | ----------- | ------------------ | -------------- |
> | FP16        | 60.38              | 15.72          |
> | GPTQ-2bit   | 130.70             | 4.14           |
> | AQLM-2bit   | 70.97              | 4.44           |
> | QuIP#-2bit  | 79.60              | 4.13           |
> | UniSVQ-2bit | 101.65             | 3.87           |
>
> UniSVQ achieves **1.68× speedup** over FP16 and outperforms all VQ baselines in both throughput and memory. GPTQ is fastest but collapses at 2-bit (PPL > 46,000 on Qwen-3-8B), making it impractical. Among methods with usable 2-bit accuracy, UniSVQ delivers the best throughput.
>
> **W4 & Q3**: The entire process requires 6 hours on an A100 for an 8B model, which is not cheap. What is the most costly part of the quantization pipeline?
>
> **Response**: 6 hours is a one-time offline cost, standard and affordable in PTQ. For reference, AQLM requires ~24 hours on comparable hardware and Quip# requires ~6 hours; fine-tuning-based PTQ methods (e.g., OmniQuant) involve similar or longer calibration times. This one-time cost amortizes quickly against long-term inference latency and GPU memory savings that accrue at every deployment.
>
> **Q1**: Is this method applicable to KV-cache quantization?
>
> **Response**: The affine-constrained quantization grid is in principle applicable to KV-cache. However, KV-cache tensors are generated dynamically at each decoding step with token-dependent distributions, so the fine-tuning strategy (Section 3.2.3) would need adaptation, e.g., via online calibration or per-head statistics. We consider this a valuable future direction.
>
> **Q2**: What are the pros and cons of applying this method to MoE models?
>
> **Response**: Each expert's weight matrix can be independently quantized with its own affine parameters. With only 20 extra parameters per matrix, overhead remains negligible even with many experts. A potential challenge is that different experts are activated at different frequencies during calibration, which may cause uneven Hessian estimation for rarely activated experts. Importance-weighted calibration could address this. We plan to explore MoE quantization in future work.

---

> > ### Author Rebuttal · Reviewer_AxuW · 2026-04-03
> >
> > Fully resolved

---

> > > ### Author Response · Authors · 2026-04-03
> > >
> > > We sincerely thank the reviewer for acknowledging that **all concerns have been fully resolved**. Given this, we would kindly ask the reviewer to consider updating the overall score to reflect this assessment, as also suggested by the acknowledgement prompt. If the reviewer still has remaining concerns or sees specific limitations that justify the current score, we would appreciate the opportunity to discuss them further before the deadline.

---

### Decision · Program_Chairs · 2026-04-30

**Decision:**

Accept (regular)

**Comment:**

This paper presents a nice method to connect and unify the simplicty of Scalar Quantization and accuracy of Vector Quantization to improve ultra low-precision 2 bit performance. All the reviewers appreciate the simplicity of the algorithm. I strongly recommend authors to follow-up on the discussions in the next revision. Overall, I beleive this paper adds lots of value to the community.